# The Alliance AMBUSH Trial: Rationale and Design

**DOI:** 10.3390/cancers14020414

**Published:** 2022-01-14

**Authors:** Anita Mahajan, Helen Shih, Marta Penas-Prado, Keith Ligon, Kenneth Aldape, Leland S. Hu, Ashlee R. Loughan, Michael R. Basso, Heather E. Leeper, Brian V. Nahed, Shannon L. Stott, Susan Geyer, Caterina Giannini, Evanthia Galanis

**Affiliations:** 1Department of Radiation Oncology, Mayo Clinic, Rochester, MN 55905, USA; 2Department of Radiation Oncology, Massachusetts General Hospital, Boston, MA 02114, USA; HSHIH@mgh.harvard.edu; 3Neuro-Oncology Branch, National Cancer Institute, National Institutes of Health, Bethesda, MD 20892, USA; marta.penas-prado@nih.gov (M.P.-P.); heather.leeper@nih.gov (H.E.L.); 4Department of Pathology, Brigham and Women’s Hospital, Boston, MA 02115, USA; Keith_Ligon@dfci.harvard.edu; 5Laboratory of Pathology, National Cancer Institute, Bethesda, MD 20892, USA; kenneth.aldape@nih.gov; 6Department of Radiology, Mayo Clinic, Phoenix, AZ 85054, USA; hu.leland@mayo.edu; 7Department of Neurology, Virginia Commonwealth University, Richmond, VA 23284, USA; ashlee.loughan@vcuhealth.org; 8Department of Psychiatry and Psychology, Mayo Clinic, Rochester, MN 55905, USA; basso.michael@mayo.edu; 9Department of Neurosurgery, Massachusetts General Hospital, Boston, MA 02114, USA; BNAHED@MGH.HARVARD.EDU; 10Center for Cancer Research, Massachusetts General Hospital, Boston, MA 02114, USA; sstott@mgh.harvard.edu; 11Department of Quantitative Health Sciences, Mayo Clinic, Rochester, MN 55905, USA; geyer.susan@mayo.edu; 12Department of Pathology, Mayo Clinic, Rochester, MN 55905, USA; giannini.caterina@mayo.edu; 13Department of Biomedical and Neuromotor Sciences, University of Bologna, 40126 Bolgona, Italy; 14Department of Oncology, Mayo Clinic, Rochester, MN 55905, USA; galanis.evanthia@mayo.edu

**Keywords:** medulloblastoma, sonic hedgehog, sonidegib, pineal parenchymal tumor, clinical trial, radiotherapy, targeted therapy

## Abstract

**Simple Summary:**

Medulloblastoma, the most common embryonal tumor in children, can also arise in older patients. Clinical studies in children with medulloblastoma have increased our understanding of molecular pathways and improved treatment strategies. We now know that medulloblastoma has at least four subtypes and each maybe best suited to specific therapies. The sonic hedgehog (SHH) pathway is altered in a significant proportion of older patients with medulloblastoma. The Alliance for Clinical Trials in Oncology cooperative group is developing the AMBUSH trial: *Comprehensive Management of Adolescent and Young Adult (AYA) and Adult Patients with Medulloblastoma or Pineal Embryonal Tumors With A Randomized Placebo Controlled Phase II Focusing on Sonic Hedgehog Pathway Inhibition in SHH Subgroup Patients (Adult & Adolescent MedulloBlastoma Using Sonic Hedgehog Trial).* The trial gives treatment directions for all patients and randomizes patients with average risk SHH-activated medulloblastoma to maintenance sonidegib, a hedgehog signaling pathway inhibitor, or placebo. This trial will establish a baseline for future trial comparison and investigate the benefit of a novel targeted agent.

**Abstract:**

Unlike medulloblastoma (MB) in children, robust prospective trials have not taken place for older patients due to the low incidence of MB in adults and adolescent and young adults (AYA). Current MB treatment paradigms for older patients have been extrapolated from the pediatric experience even though questions exist about the applicability of these approaches. Clinical and molecular classification of MB now provides better prognostication and is being incorporated in pediatric therapeutic trials. It has been established that genomic alterations leading to activation of the sonic hedgehog (SHH) pathway occur in approximately 60% of MB in patients over the age of 16 years. Within this cohort, protein patched homolog (PTCH) and smoothened (SMO) mutations are commonly found. Among patients whose tumors harbor the SHH molecular signature, it is estimated that over 80% of patients could respond to SHH pathway inhibitors. Given the advances in the understanding of molecular subgroups and the lack of robust clinical data for adult/AYA MB, the Alliance for Clinical Trial in Oncology group developed the AMBUSH trial: Comprehensive Management of AYA and Adult Patients with Medulloblastoma or Pineal Embryonal Tumors with a Randomized Placebo Controlled Phase II Focusing on Sonic Hedgehog Pathway Inhibition in SHH Subgroup Patients (Adult & Adolescent MedulloBlastoma Using Sonic Hedgehog Trial). This trial will enroll patients 18 years of age or older with MB (any molecular subgroup and risk stratification) or pineal embryonal tumor. Patients will be assigned to one of three cohorts: (1) average risk non-SHH-MB, (2) average risk SHH-MB, and (3) high risk MB or pineal embryonal tumors. All patients will receive protocol-directed comprehensive treatment with radiation therapy and chemotherapy. Patients with SHH-MB in cohort 1 will be randomized to a smoothened inhibitor or placebo as maintenance therapy for one year.

## 1. Introduction

Medulloblastoma (MB) is the most common malignant brain tumor of childhood, yet it accounts for only 1% of adult brain tumors [1,2]. Despite this low incidence, 30% of all medulloblastoma cases are diagnosed in individuals between age 15 and 39, patients who often have limited access to clinical trials. Over the past several decades, sequential cooperative group non-infant pediatric MB studies have resulted in better outcomes with a multidisciplinary approach incorporating surgery, comprehensive radiation, and multi-agent chemotherapy.

After complete staging evaluation patients are found to either average- or high-risk groups. Patients with average-risk disease have neither residual disease (i.e., <1.5cm^2^) nor metastatic disease, (i.e., M0). All others are staged into the high-risk group. In addition, it has been noted the presence of anaplasia or *MYC* amplification even with a complete resection and M0 disease are associated with poorer outcomes. These patients are often assigned to the high-risk group. Currently, based on several multi-institutional trials, pediatric patients with average-risk and high-risk MB have an estimated 5-year overall survival of 85% and 50–75%, respectively [3,4,5,6,7,8].

Unfortunately, due to the low incidence of the disease adult MB treatment paradigms have been challenging to establish. Pediatric data have been extrapolated to the older population, but concerns exist regarding the differences in tumor biology and treatment tolerance. Adult MB may represent a different biological spectrum than children, but many characteristics are also shared thus there is uncertainty of the validity of paradigms from the pediatric experience in an older population [9,10,11,12].

Over the last 10 years MB has become recognized as a heterogeneous disease with important biologic, molecular and clinical risk factors. The basic subgroups are Wingless/Integrated-activated (WNT), Sonic Hedgehog-activated (SHH), Group 3 and Group 4, each with a characteristic age spectrum and disease outcome [13,14,15,16]. The distribution of molecular subgroups in adults and children varies: 60–70% of adult MB have an aberrant activation of the SHH pathway, as compared to only 30% in children [17,18,19]. The subgroup 3 MB represent about 25% of cases in children, but are not common in adults, whereas the WNT and Group 4 tumors appear to have a worse outcome in adults than in children based on retrospective data. Finally, the frequency of high-risk metastatic disease (i.e., disseminated outside of the posterior fossa and/or systemically) at initial presentation is lower in adults than in children [20,21,22].

The Alliance for Clinical Trial in Oncology group is planning the AMBUSH trial: Comprehensive Management of Adolescent and Young Adult (AYA) and Adult Patients with Medulloblastoma or Pineal Embryonal Tumors with a Randomized Placebo Controlled Phase II Focusing on Sonic Hedgehog Pathway Inhibition in SHH Subgroup Patients (Adult & Adolescent MedulloBlastoma Using Sonic Hedgehog Trial), A072001. This will be a comprehensive trial that includes all patients 18 and older with medulloblastoma or pineal embryonal tumors divided into three cohorts: (1) average-risk non-SHH-MB, (2) average-risk SHH-MB, and (3) high-risk MB or pineal embryonal tumors. All eligible patients will receive protocol directed comprehensive treatment with radiation therapy and chemotherapy. Patients with SHH-MB in cohort 1 will be randomized to a smoothened inhibitor or placebo as maintenance therapy for one year. This protocol once completed will provide robust guidance for the management of medulloblastoma and pineal embryonal tumors in the non-pediatric population.

## 2. Classification of Medulloblastoma

The 2021 WHO classification for MB requires distinction of SHH- and WNT MB from non-WNT/non-SHH tumors. Pediatric MB paradigms now incorporate biologic in addition to clinical prognostic factors with the following goals: (1) toxicity reduction for very low risk disease (WNT+, M0) by reduction in RT dose to the CSI and the primary site, (2) improved tumor control strategies for patients with high-risk disease (metastatic disease, anaplastic MB, myc amplification, Group 3 and 4 or non WNT/SHH) using high dose chemotherapy and/or varying systemic agents. NCT00392327, NCT01878617, and NCT02724579).

In this study, we will incorporate upfront classification of MB subtype by integral centrally performed molecular analysis including immunohistochemistry, copy number assessment, and DNA sequencing for cohort assignment and randomization. Prospective genome-wide DNA methylation profiling will also be performed to fully characterize each tumor and confirm classification before randomization.

## 3. Rationale to Include Non-SHH-MB, High-Risk MB, and Pineal Tumors

The intent of this unique trial is to leverage the opportunity and gain knowledge on all other populations of adult/AYA patients with MB (non-SHH and/or high risk) and pineal embryonal tumors since the treatment platform for all these subgroups is like the one proposed for the randomized group. Brandes et al., reported on 36 adult patients, 11 of whom had M+ disease, treated over a 12-year span. In this study the PFS for adult patients was similar to similarly staged pediatric patients with high-risk disease having a poorer outcome [12]. Herrlinger et al., reported on a retrospective study of 36 adult patients treated over 22 years with MB and supratentorial primitive neuroectodermal tumors and noted that chemotherapy appeared to prolong survival [23]. The authors concluded more study was required and future efforts may build on this experience with the addition of new knowledge from ongoing and completed pediatric trials.

Because the natural history and treatment paradigms for pineal parenchymal tumors of intermediate differentiation (PPTID) vary from poorly/undifferentiated pineal parenchymal tumors, patients with PPTID will not be eligible [24]. Patients with non-SHH-MB, high-risk MB or pineal embryonal tumors will be enrolled on non-randomized arms and treated in a prescribed manner to determine objective outcomes since robust prospective data are not available for these patients. All patients with pineal region embryonal tumors will undergo full molecular characterization to establish prospective information for future analysis.

Non-randomized study objectives for this group will be to characterize clinical outcomes for pineal embryonal tumors and high-risk MB with comprehensive radiotherapy followed by adjuvant chemotherapy in a prospective trial setting. Further, in this rare disease setting of adult MB and pineal embryonal tumors patients, establishing a prospectively and rigorously collected set of data on a cohort treated with the current standard of care facilitates its potential use as external or ‘synthetic’ controls when evaluating experimental regimens in this same rare disease setting.

## 4. Backbone of AMBUSH Therapeutic Strategy

### 4.1. Surgery

Surgical intervention for posterior fossa tumors is important to re-establish CSF flow when there is obstructive hydrocephalus, to provide tissue diagnosis, and to decompress in the setting of mass effect. It has been noted in several MB studies that a complete resection has prognostic implication [8,25,26]. In our study, the degree of resection will be used for cohort assignment, where those with residual tumor greater than 1.5 cm^2^ on cross sectional measurement will be assigned to the high-risk arm. The degree of resection will be evaluated as an independent factor on disease outcomes including local recurrence and disease control. We will also assess for drop metastases along the spinal axis to further stratify eligibility and survival outcomes.

### 4.2. Radiotherapy

After studying post mortem MB failure patterns, Paterson and Farr recommended irradiating the brain and spinal cord in one undivided volume in the 1940s [27]. Their publication in 1953 reported the outcomes of 22 children and five adults in whom 65% had 3-year overall survival (OS). Craniospinal irradiation (CSI) has become part of the standard of care for treatment of MB with curative intent. The CSI dose in this early study was 3000–3500 roentgen to the spine and a maximum of 5000 roentgen to the cerebellum, with the limit set by tolerance of the normal tissues. Of note, a pediatric study (POG8631/CCG923) randomized pediatric patients with average risk MB to radiation therapy alone of 23.4 Gy CSI vs. 36 Gy CSI, both with a posterior fossa boost to 54 Gy, and reported an 8-year event-free survival (EFS) of 52% vs. 67%, respectively [8]. The standard CSI dose for average risk MB CSI remained 36 Gy when used without chemotherapy. Currently for patients 18 years and younger with average-risk disease, 23.4 Gy CSI with adjuvant chemotherapy is the standard of care with similar outcomes to 36Gy CSI without chemotherapy.

In adults, data for the use of lower dose CSI with chemotherapy is limited. Friedrich et al. reported a 4-year EFS of 68% and 4-year OS of 89% with no difference between 23.4 Gy CSI (*n* = 9) and 35.2 Gy CSI (*n* = 47) on the HIT 2000 prospective observational study [28]. Massimino et al., reported on 44 adults with non-metastatic MB treated at two European centers treated with <36 Gy CSI and chemotherapy. Thirty-six patients received 23.4 Gy CSI and eight patients had 30.6 Gy CSI in addition to a posterior fossa boost and chemotherapy based on pediatric protocols at each institution. The 5- and 10-year progression-free survival (PFS) was 82.2% and 78.5%, respectively. The OS rates were 89% and 75.2% at 5 and 10 years [26]. Similarly, Majd et al., published a large retrospective single institution study of adult MB in which 16 of 53 patients with standard-risk MB received <30 Gy CSI and no difference in outcome was noted [29].

A transition from a boost to the entire posterior fossa to a conformal boost of the primary site surgical bed has taken place as more conformal radiotherapy techniques have become available. The benefit has been lower dose to the cochlea, brainstem, spinal cord, and supratentorial brain. Michalski et al., reported the results of COG ACNS 0331 that randomized 464 children with average-risk MB to whole posterior fossa vs. involved field boosts with 5-year EFS 80.5% vs. 82.5%, respectively [7].

Based on these data and the investigators pooled experiences, in this study, 23.4 Gy CSI followed by a 30.6 Gy boost to the primary site will be used for all patients with average-risk disease. For all high-risk patients, the CSI dose will be 36 Gy followed by boost (s) to the primary (54 Gy) and metastatic (45–54 Gy) sites. All patients will receive adjuvant chemotherapy and high-risk patients will receive concurrent vincristine during CSI.

### 4.3. Chemotherapy

Adults do not tolerate chemotherapy as well as children as noted by several groups. Adult patients experience prolonged neutropenia, increased neuropathy, and require dose modifications or chemotherapy cessation due to toxicities [28,30,31]. There is no agreement on the optimal chemotherapy regimen for medulloblastoma in adults; chemotherapy regimens frequently used in clinical trials for average- and high-risk disease are summarized in Table 1 and Table 2. The proposed regimen using cisplatin, vincristine, and cyclophosphamide is the result of an in-depth discussion within the neuro-oncology committee of Alliance that represents most major academic centers in the US and the NCI Connect adult MB workshop [11]. Our hypothesis is that the use of an SHH inhibitor will allow chemotherapy de-escalation in patients with the appropriate molecular characteristics.

For high-risk disease, since their outcomes are less well known and tend to be worse, the AMBUSH study will maintain the use of vincristine during radiotherapy every two weeks and adjuvant chemotherapy as noted below in Table 2.

### 4.4. Targeted Therapy

The Hedgehog (Hh) pathway is essential in embryonic development and tumorigenesis. SHH is an extracellular Hh protein that is involved in nervous system development. The normal SHH pathway includes transmembrane protein, PTCH which constitutively inhibits SMO, another transmembrane protein, from internalization. If SHH binds with PTCH then, PTCH and SMO are internalized, resulting in decoupling of GLI (glioma-associated oncogene) from SUFU (negative regulator suppressor of fused). GLI then triggers transcription of hedgehog factors involved with growth. PTCH or SMO mutations can result in unopposed Hh pathway activation resulting in unregulated growth [32,33].

It has been established that SHH pathway mutations occur in approximately 60% of MB in patients over the age of 16, i.e., the AYA and adult populations. Unlike younger patients where GLI, SUFU, MYCN, and TP53 mutations are noted, MB in older patients is commonly associated with *PTCH* and S*MO* mutations [17,18,19]. With this molecular signature, over 80% of patients could respond to SHH pathway inhibitors [34]. Kieran et al., evaluated sonidegib in a phase II study for 60 pediatric patients with relapsed solid tumors (39 MB) and 16 adults with relapsed MB. In this study, five of the 10 patients with an activated Hh pathway (SHH-MB) demonstrated a complete or partial response. None of the 50 patients with a negative Hh signature responded [35]. Given this background and promising preliminary results, a randomized trial to evaluate the effect of SHH pathway inhibition in AYA and adult MB with SHH mutations is needed.

The primary objective in the AMBUSH trial will be to evaluate the use of sonidegib, a SMO inhibitor randomized to placebo as maintenance after completion of CSI and chemotherapy in patients with average-risk SHH-MB, who represent an enriched population of MB with SHH pathway mutations at the level or upstream of SMO.

Based on pharmacokinetic studies it is known that sonidegib has an estimated half-life of 28 days and reaches a steady state level after 4 months. The BOLT study studied sonidegib for patients (median age 65) with advanced or metastatic basal cell carcinoma [36]. Patients were randomized to sonidegib at 200 mg (*n* = 79) vs. 800 mg (*n* = 151) daily and had a median duration of exposure of 11 months and 6.6 months, respectively. Overall, discontinuation was due to a variety of issues including 40% due to adverse events, 16% due to progressive disease, 34% due to physician or patient choice. In the whole group, the median duration of response was greater than 2 years and disease control rate was over 90%.

In the Kieran study, the median duration of exposure ranged from 34–511 days (median 97 days). Six out of 16 patients experienced grade 3/4 toxicities: 5 CPK, 1 AST, 2 ALT elevations. The authors recommended 680 mg/m^2^ as a single daily dose [35]. For our study we have chosen a flat dose of 600 mg/day based on the data summarized above, toxicity profiles, and the biodistribution differences between the CNS and skin. In addition, we have institutional experience (MGH-Shih) for tolerance and efficacy.

## 5. Trial Design

The overall schema of the AMBUSH trial is shown in Figure 1. Patients will be registered and the diagnosis of either MB or pineal parenchymal tumor will be confirmed by initial central pathology review. Following full clinical review, immunohistochemistry, DNA sequencing, copy number assessment, and genome-wide DNA methylation studies, patients will be assigned to either cohort 1 or 2 and their eligibility for randomization will be determined. Cohort 1 will receive CSI 23.4 Gy CSI followed by a tumor bed boost, whereas the higher-risk cohort 2 will receive CSI 36 Gy. Only patients with average-risk SHH-MB will be randomized to sonidegib or placebo as maintenance after completing chemotherapy.

While the nonrandomized arms will be analyzed in a descriptive manner, the primary analysis will be on those average-risk SHH-MB patients randomized to SHH pathway inhibition versus placebo. Here, this study design will provide 80% power to detect a true hazard ratio of 0.45 for PFS (corresponding to 5-year PFS rate of 70% vs. 85%), with a one-sided Type I error constraint of 0.10 and an interim analysis for futility conducted after 50% of expected events are observed.

## 6. Objectives

The primary objective is to evaluate the ability of SHH pathway inhibition maintenance therapy to improve progression-free survival (PFS), compared to placebo, in average risk patients with SHH-MB. Secondary objectives will include evaluating PFS in average-risk non-SHH-MB patients, high-risk MB, and pineal embryonal tumor patients with the aim to describe other clinical outcomes in these less-common subtypes.

## 7. Accrual Goal

A total of 108 patients will be accrued for randomization to the average-risk SHH-MB arm. The other groups will be open to accrual for the full duration of accrual to the randomized arms. It is estimated that a total of 20–40 patients will be registered to each of the non-randomized arms.

## 8. Translational Research

This trial presents a unique and highly valuable opportunity to understand the full spectrum of patient-centered outcomes, disease correlatives and treatment related morbidities. Patients will undergo specific evaluations at baseline and subsequent visits to facilitate a greater understanding of patient-centered outcomes. The planned tests and observations are summarized in Table 3.

### 8.1. Neurocognitive Function

Most people treated for MB survive at least 10 years [37,38]. Much of the literature concerning neurocognitive functioning among MB survivors involves children or individuals treated during childhood. Few studies of adult MB survivors have been conducted. Harrison et al., conducted a cross-sectional study of 27 patients treated for MB during early adulthood in which 25–30% of patients displayed impaired executive function and new-learning performance that correlated with the amount of chemotherapy cycles received [39]. Dirven et al. reported on 28 patients treated with radiation and chemotherapy. Working memory and attention were generally impaired after treatment and remained compromised during the follow-up period. In contrast, executive function and information processing speed improved over the follow-up period [40].

These investigations are limited by relatively small sample sizes, but they are representative of the limited literature concerning MB in adults. They suggest that treatment of MB corresponds with attenuated executive function, working memory, and new learning. Patients enrolled on this study will have serial neurocognitive testing to determine frequency and extent of neurologic compromise and evaluate the contributing factors.

### 8.2. Health Related Quality of Life

There is very little information about the health-related quality of life outcomes for older patients treated for MB or pineal region tumors. The current trial presents a significant opportunity to assess health-related quality of life prospectively and systematically in adults receiving chemoradiation following surgery and for patients with SHH-mutated tumors randomized to the sonidegib arm with this addition of a SMO inhibitor added to standard of care chemoradiation. The EORTC QLQ-30 and QLQ-BN-20 questionnaires were chosen for the current study to harmonize with companion studies and take 15–20 min to complete. These longitudinal patient-reported outcome assessments will coincide with imaging assessment visits to minimize burden to the patient. Additionally, PRO-CTCAE will also be assessed during treatment. The study of patient-reported symptom burden and health-related quality of life, which encompasses the patients’ perception of their physical, psychological, and social well-being, are critically important in better understanding the implications of the proposed treatment on living with this disease and through its treatment, in both the acute and long-term settings, above and beyond the disease-centered outcomes of PFS and OS.

### 8.3. Neuroendocrine, Auditory, and Visual Functions

MB treatment can affect normal neuroendocrine, auditory, and visual functions with the use of CSI and chemotherapy such as cisplatinum [41,42,43,44,45,46]. In this study we will collect baseline and subsequent functions to objectively evaluate subsequent morbidity in these patient cohorts. Dose–volume relationships with radiotherapy and chemotherapy will be determined.

### 8.4. Radiotherapy Modality

Retrospective studies have suggested improved tolerance of radiotherapy and subsequent chemotherapy with the use of proton therapy [47]. In children there is evidence of reduced auditory, endocrine, and neurocognitive toxicities when proton therapy is used in comparison to X-ray based therapies [43,44,45,46,47,48]. In this study we will stratify patients in the randomized arm based on whether they are treated with proton beam therapy or X-ray based radiation and note radiotherapy modality in all patients to prospectively assess whether findings noted in the studies above hold true in the adult population as well.

### 8.5. Effect of Memantine

Memantine has been reported to reduce neurocognitive morbidity in patients receiving whole brain radiotherapy for brain metastasis and now has become a standard of care [49,50,51]. Similarly, memantine is now being studied in children to reduce radiation related toxicities (NCT 03194906, NCT 04217694). In this study, use of memantine during radiotherapy will be used as a stratification factor for randomization and will be noted in all patients to determine whether neurocognitive outcomes or tumor control are affected by its use.

### 8.6. CSF and Plasma Banking

CSF and plasma will be obtained in a serial fashion for banking. Studies are planned to determine whether liquid biopsy markers such as circulating tumor DNA, extracellular vesicles, or other markers can predict tumor recurrence or other high-risk features.

### 8.7. Comparison to Contemporaneous Trials

The AMBUSH study will overlap with an EORTC adult MB studies in Europe. Though the studies are not identical, there will be a unique opportunity to compare outcomes in the average-risk patients. Between the two studies, even though each cohort is relatively small, we will be simultaneously answering whether concurrent or maintenance SMO inhibition is tolerated and/or effective. In addition, we will have prospective data collected that could be merged to evaluate other outcomes such as hearing loss, quality of life or cognition. Similarly, this will provide a unique opportunity to compare findings with ongoing pediatric trials that are building on MB subtype tailored treatment algorithms and toxicity reduction efforts.

## 9. Conclusions

This trial aims to set a standard of care for treatment of AYA and adult MB. It incorporates a strategic and innovative molecular targeted agent for the most common type of MB in this patient cohort (SHH). We will be able to prospectively determine if this cohort of older patients has similar outcomes to the pediatric population when treated with CSI and adjuvant chemotherapy. The trial will include Health related quality of life (HRQOL), neurocognitive, endocrine, auditory, and visual assessments to evaluate full cohort outcomes as well as relationships with normal tissue dose distributions.

## Figures and Tables

**Figure 1 cancers-14-00414-f001:**
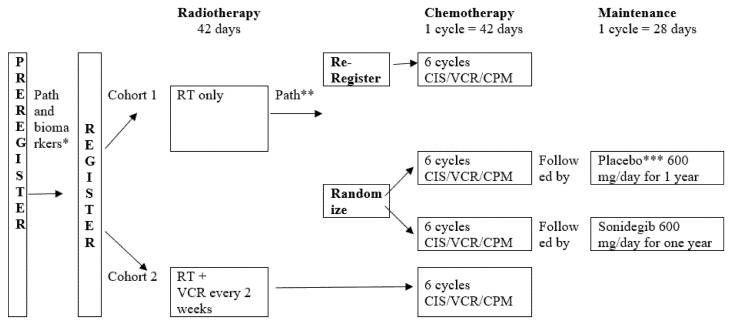
Overall schema of the AMBUSH trial (A072001) for patients 18 years of age at diagnosis or older with medulloblastoma or pineal embryonal tumors. * Preregister on study and begin clinical staging, RT plan preparation. Central pathology review and biomarker assays: IHC, DNA seq, CNA and clinical staging (postoperative brain MRI, spinal MRI, CSF analysis) required to assign to cohort: Cohort 1: average risk MB (non-SHH or SHH arms); Cohort 2: high risk MB or any PET. ** Final integrated pathological diagnosis including results of DNA methylation for MB subgroup confirmation (SHH vs. Non-SHH—Group 4, WNT). *** Patients receiving placebo who relapse will be allowed to crossover. Abbreviations: RT: radiotherapy; MB: medulloblastoma; IHC: immunohistochemistry; DNA seq: DNA sequencing; CNA: copy number assessment; SHH: sonic hedgehog; yo: years old; GTR: gross total resection; NTR: near total resection; M0: no metastatic disease; M1–3: any metastatic disease; CSI: craniospinal irradiation; q2wk: every other week; CIS: cisplatin; VCR: vincristine; CPM: cyclophosphamide; PS: performance status; XRT: x-ray radiotherapy; PRT: proton radiotherapy.

**Table 1 cancers-14-00414-t001:** Chemotherapy agents that are used in the treatment in average-risk medulloblastoma with final AMBUSH regimen in comparison to other recently reported and ongoing trials.

Tx Phase	CHEMO	AMBUSH	A9961	ACNS0331	NOA-7	SJMB03	SJMB12
	Number of cycles, (doses mg/m^2^)	**6 cycles as** **tolerated**	A 8 (ccnu/cis/vcr) vs. B 8 (cpm/cis/vcr)	A 6 (ccnu/cis/vcr) B 3 (cpm/vcr) AAB	ADULT	A 4 (vcr/cpm/cis)	A 4 (vcr/cpm/cis)
RT	VCR	**None**	8 (1.5)	6 (1.5)	3 (1.5)	none	none
MAINTENANCE	VCR	**6 (1.5 × 2)** **max (2) mg**	8 (1.5 × 3)	A 6 (1.5 × 3) + B 3 (1.5 × 2)	6 (1.5 × 2)	4 (1.0 × 2)	4 (1.0 × 2)
	CIS	**6 (75)**	8 (75)	A 6 (75)	6 (70)	4 (75)	4 (75)
	CCNU	**none**	8 (75)	A 6 (75)	6 (75)	none	none
	CPM	**6 (1000 × 2)**	8 (1000 × 2)	B 3 (1000 × 2)	none	4 (2000 × 2)	4 (1500 × 2)
	VP16	**none**	none	none	none	none	none
Comments		**6 cycles if** **tolerated**	reg A = B	std COG	tolerated	stem cell rescue ×4	SHH: vismodegib maintenance

Abbreviations: Tx: Treatment; CHEMO: chemotherapy; VCR: vincristine; CIS: cisplatin; CPM: cyclophosphamide; VP16: etoposide. COG: Children’s Oncology Group; Trial Sponsors: A9961: Pediatric Oncology Group; ACNS033: COG; NOA-7: German Neuro-Oncology Working Group; SJMB03 and SJMB12: St Jude Children’s Research Hospital.

**Table 2 cancers-14-00414-t002:** Chemotherapy agents that are used in the treatment in high-risk risk MB with final AMBUSH regimen in comparison to other recently reported and ongoing trials.

Tx Phase	CHEMO	AMBUSH	ACNS 0332	SJMB03	SJMB12- SHH	SJMB12
	All doses mg/m^2^	**6 cycles as** **tolerated**	6 cycles	4 (vcr/cpm/cis)	4 (vcr/cpm/cis)	A 4 (vcr/cpm/cis) B 3 (paclitaxel/gem) AABAABB
RT	VCR	**6 (1.5)** **q2 wk**	6 (1.5) weekly	none	none	none
	Carboplatin	**none**	30 (35) daily	none	none	none
MAINTENANCE	VCR	**6 (1.5 × 2)**	6 (1.5 × 2)	4 (1.0 × 2)	4 (1.0 × 2)	4 (1.0 × 2)
	CIS	**6 (75)**	6 (75)	4 (75)	4 (75)	4 (75)
	CPM	**6 (1000 × 2)**	6 (1000 × 2)	4 (2000 × 2)	4 (1500 × 2)	4 (1500 × 2)
	Paclitaxel	**none**	none	none	none	3 (600 × 2)
	Gemcitibine	**none**	none	none	none	3 (1250 × 2)
COMMENTS		**6 cycles** **if tolerated**	daily carbo randomized	4 × stem cell rescue	vismodegib maintenance	intermediate, HR Non-SHH

Abbreviations: Tx: treatment; CHEMO: chemotherapy; VCR: vincristine; CIS: cisplatin; CPM: cyclophosphamide; VP16: etoposide; carbo: carboplatin; gem: gemcitabine; COG: Children’s Oncology Group; SHH: sonic hedgehog; Trial Sponsors: ACNS032: COG; SJMB03 and SJMB12: St Jude Children’s Research Hospital.

**Table 3 cancers-14-00414-t003:** Baseline and subsequent tests and observations for patients enrolled on the AMBUSH Study.

Test or Observation	Baseline	Weekly during RT	Prior to Chemo	Chemo × 4 Cycles	Pre Sonidegib/Placebo	Monthly	Annually Post Completions				
**Approximate Wk,** **Wk 0 = RT start**	**Wk**	**Wk**	**Wk**	Wk	Wk	Wk	Wk 52				
	−4 to 2	0 to 6	9 to 10	10 to 26	26 to 30	30 to 82					
**Tests and Observations**											
History and Physical	pre RT	x	x	x	x	each visit	x				
Weight (each visit)	pre RT	x	x	prn	x	x	x				
Hearing	x			x	x		y 1, 3, 5				
Vision	x				x		y 1, 3, 5				
CompNeurocog	x						y 1, 3, 5				
Short Neurocog	x				x		x	x	x	x	x
QOL/PRO-CTCAE	x	x	x	x	x		y 1, 3, 5				
**Laboratory Studies**											
CBC	pre RT	x	x	x	x	x	x	x	x	x	x
Blood Chemistries	x		x	prn	x	x	x	x	x	x	x
Endocrine	x				x		y 1, 3, 5				
Creatine Kinase	x		x	x	x	x	x	x	x	x	x
BUN/Creatinine	x		x	x (1)	x	x	x	x	x	x	x
**Staging**											
Tumor Imaging	pre RT		x	post cycle 2	x						
Research Studies for Banking	tumor, CSF, plasma		plasma		plasma						
	plasma and CSF if Recurrence										

n.b. x denotes when the test or observation will be required on the protocol. Abbreviations: Wk: week; RT: radiotherapy; chemo: chemotherapy; Comp Neurocog: comprehensive neurocognitive testing; Short Neurocog: Short neurocognitive testing; QOL: quality of life; PRO: patient reported outcomes; CBC: complete blood count; BUN: blood urea nitrogen; CSF: cerebrospinal fluid; prn: as needed; y: year.

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
