# Peer review of "The Alliance AMBUSH Trial: Rationale and Design"

_cancers, 2022, doi:10.3390/cancers14020414_

Round 1

Reviewer 1 Report

This article presents a randomized phase 2 protocol with both a protocol for the management of common adult medulloblastomas and a randomization forstandard risks SHH activated medulloblastomas between adjuvant chemotherapy plus placebo and adjuvant chemotherapyand anti-SHH. As the authors specify, there is a clinical, pathological and prognostic specificity of medulloblastomas in adults which deserves specific consideration.

In comments:

1 / These specificities and the remaining gray areas for adults medulloblastomas are well introduced in the different chapters even if a focus on the existing doubts on the relevance of the Chang classification and the prognostic roles of the residue and the presence of metastases in adults could have been more detailed (1-2) .

2 / We submit to the authors the potential place of the PNET HR + 5 protocol in chemotherapy protocols, which recently showed a good response profile for high-risk pediatric patients by offering neo-adjuvant chemotherapy including 2 high dose thiotepa followed by radiation therapy and adjuvant temodal (3).

3 / As specified by the authors, recurrence of adult medulloblastomas occurs later than in the pediatric population with a median of 6.6 years (4). What follow-up is planned after 5 years to ensure the diagnosis and treament these recurrences?

4 / The authors have perfectly prepared the aspects of neurocognitive monitoring in the medium term. What will this follow-up be beyond the 5 years of the main objective.

  1. Alba a. Brandes, M.D., the treatment of adults with medulloblastoma: a prospective study Int. J. Radiation Oncology Biol. Phys., Vol. 57, No. 3, pp. 755–761, 2003
  2. Ulrich Herrlinge, M.D. Adult medulloblastoma Prognostic factors and response to therapy at diagnosis and at relapse J Neurol (2005) 252 : 291–299
  3. NCT00936156
  4. Majd, N.K., et al., Clinical characterization of adult medulloblastoma and the effect of first-line therapies on outcome; The MD Anderson 414 Cancer Center experience. Neurooncol Adv, 2021. 3(1): p. vdab079.

Author Response

Thank you for your insightful and helpful comments. 

Comment 1 and 2. I have added the references as suggested and added content in the section supporting the need to include high risk patients and this will lay the ground work for future studies with other treatment approaches that will be learned from pediatric studies. 

Comment 3 and 4: Yes this is an important issues.  I do hope that additional research support will be requested and become available for longer term follow up.  We will be following all patients for vital status after 5 years, but the current protocol funding will need to expanded for additional late endpoints.  

I appreciate your input

Regards

Anita Mahajan

Reviewer 2 Report

The authors present one of two planned randomized clinical trials for newly diagnosed medulloblastomas (MB) in AYA and adults, a rare tumor manifestation without an uniformly accepted standard of care. Their own trial will run in the US, whereas the EORTC 1634 trial is a worldwide endeavor. Both trials will explore the role of SMO inhibitors in SHH MBs and a reduced dose of radiation in postpubertal (EORTC 1634) and patients ≥ 18 years of age (ALLIANCE trial) respectively. 

Endpoints will focus on benefits and toxicities of the experimental regimens in these age groups. Substudies are planned in both trials.

Unfortunately, the two very similar, well designed, randomized trials will be running in parallel with only a few patients each, 108 patients in the EORTC 1634 and 108 patients in the ALLIANCE trial, which is a pity in this rare tumor manifestation. Both trials aim to set a standard of care for AYA and adults with MBs. Whether the study results can be compared retrospectively remains to be seen, at least endpoints have been agreed upon by both trial coordinators.

There is no doubt about the need for such a study and it should  be carried out worldwide.

Since both studies (EORTC 1634 and ALLIANCE) are  well advanced in their planning, there seems to be no possibility of a merger. This fact should be explained in the paper in more detail (section 8.7) also in terms of how to define a standard of care based on study results of the two similar, but not identical trials (e.g. duration of SHH inhibitor).

Author Response

Thank you for the incisive review.  Your thoughts about the accumulated knowledge from the PERSOMED and AMBUSH trials are very good.  I do believe there will be outcomes that will be complimentary and we may end up having more knowledge between the two trials than if we had identical studies, for instance whether maintenance vs concurrent SHH inhibition is associated with improved outcomes and relative tolerance.  Within 5 years, hopefully, we will have a better understanding of the role of these targeted agents.  I have added a statement to reflect these thoughts in section 8.7 as suggested.